# Canine Fecal Microbiota Transplantation: Current Application and Possible Mechanisms

**DOI:** 10.3390/vetsci9080396

**Published:** 2022-07-30

**Authors:** Maimaiti Tuniyazi, Xiaoyu Hu, Yunhe Fu, Naisheng Zhang

**Affiliations:** College of Veterinary Medicine, Jilin University, Changchun 130062, China; mmttn18@mails.jlu.edu.cn (M.T.); huxiaoyu99@mails.jlu.edu.cn (X.H.)

**Keywords:** canine, fecal microbiota transplantation, treatment, mechanism

## Abstract

**Simple Summary:**

Fecal microbiota transplantation (FMT) is a newly adapted therapeutic approach in dogs. Although FMT produced a promising effect with little adverse events when treating various canine gastrointestinal disorders, safety concerns and lack of understanding regarding its therapeutic mechanisms are the main reasons limiting its wider application. Therefore, in this paper, we describe the current application and efficacy of canine FMT, as well as possible mechanisms that may be involved in the treatment process. In addition, we also discuss the future prospective of canine FMT regarding selecting donor dogs more efficiently and safely, choosing and pretreating recipient dogs to increase FMT efficacy, choosing more efficient routes of administration and stool storage, as well as its potential applications beyond gastrointestinal issues, including behavior modification and obesity treatment.

**Abstract:**

Fecal microbiota transplantation (FMT) is an emerging therapeutic option for a variety of diseases, and is characterized as the transfer of fecal microorganisms from a healthy donor into the intestinal tract of a diseased recipient. In human clinics, FMT has been used for treating diseases for decades, with promising results. In recent years, veterinary specialists adapted FMT in canine patients; however, compared to humans, canine FMT is more inclined towards research purposes than practical applications in most cases, due to safety concerns. Therefore, in order to facilitate the application of fecal transplant therapy in dogs, in this paper, we review recent applications of FMT in canine clinical treatments, as well as possible mechanisms that are involved in the process of the therapeutic effect of FMT. More research is needed to explore more effective and safer approaches for conducting FMT in dogs.

## 1. Introduction

Previously, microbiomes were considered as pathogens that lead to diseases [1]; however, it is now evident that they are an important part of the human and animal body, and play a crucial role in host physiology [2,3,4]. Recent advances in culture-independent DNA-sequencing technology (i.e., 16S rRNA sequencing) and data analysis methods revealed that every part of a dog’s body, such as the oral and nasal cavity [5,6], skin [7,8], and the gastrointestinal [9,10], respiratory [11,12,13], urinary [14], and reproductive [15,16,17] tracts, harbor certain types of microbiota.

The word ‘microbiota’ refers to all microorganisms, including bacteria, fungi, and viruses. Among them, bacteria are the most widely and deeply studied. The gut microbiome is by far the most diverse microbial community in humans as well as animals. Considering various pH levels, intestinal mobility, oxygen tension, and nutritional availability, the number of bacteria varies along the gastrointestinal tube of dogs [18]. For example, *Clostridiales* predominates in the duodenum (40% of the clones) and jejunum (39%), and are highly abundant in the ileum (25%) and colon (26%), while *Fusobacteriales* and *Bacteroidales* are the most abundant bacterial orders in the ileum (33%) and colon (30%). *Enterobacteriales* are more commonly observed in the small intestine than in the colon, and *Lactobacillales* commonly occurs in all parts of the intestine [9].

The mammalian (i.e., canine) gut begins colonization by microorganisms during birth, and is shaped by the delivery method [19]; the composition of gut microbiota influences the development of the immune system [20,21].

The canine gut microbiome contributes toward food digestion and nutrient absorption for host energy production [22], and is closely related to gastrointestinal diseases, such as colitis [23], inflammatory bowel disease (IBD) [24], *Clostridium difficile* infection (CDI) [24,25], and diarrhea [26,27]. However, recent studies have shown that gut microbiota can interact with other organs, and be the main influencing factor in health and disease, beyond the intestine alone, due to its role in host mental health and behavior [28,29].

The gut microbiome is an example of a very complex biological ecosystem. The canine gut microbiome is composed of bacteria, archaea, fungi, protozoa, yeast, viruses, and parasites, although its composition is hard to describe completely because, while it has uniformity among different groups of dogs, it also has its own unique differences between individuals [30]. These kinds of individual and group interactions also result in microbial communication that changes and maintains its relative balance; therefore, we are only able to describe the temporal dynamics of the gut microbiome [31]. In dogs, *Firmicutes*, *Bacteroidetes*, and *Proteobacteria* are the most abundant microbiota, and remain relatively more stable in the healthy gut [32]. The relative abundance of gut microbiota and their composition is vital for the effective defensive role against pathogens, and in various metabolic pathways [33].

The gut microbiome of dogs is an ecological system that keeps changing and evolving from the beginning of life to the end. Generally speaking, a state of eubiosis in the canine gut is characterized by *Firmicutes*- and *Bacteroidetes*-dominated microbial profiles [18,34,35]. It is worth mentioning the crosstalk between the canine gut microbiome and the immune system, which is crucial to overall wellbeing. It allows the body to recognize good bacteria, while protecting the body from being attacked by opportunistic bacteria that cause infections.

Fecal microbiota transplantation (FMT), which aims to restore a healthy gut microbiome after disruption by antibiotic treatment, pathogenic invasion, or dietary change, was first proposed at the 4th century AD, and has been widely studied after the US Food and Drug Administration approved its usage for the treatment of CDI in 2013 [36]. Yet, our understanding of FMT is far from sufficient, especially in dogs. Recently, FMT has been used as a therapeutic option in canine clinical treatments. Strong evidence suggests that FMT, which is characterized as transferring a suspension of fecal microorganisms from a healthy donor dog into the intestinal tract of the recipient dog, can help restore the microbial balance of a dysbiosis gut, which can result from a specific disease associated with intestinal microbiome disruption [37].

Despite numerous evidence supporting the beneficial effect of FMT, safety concerns and lack of proper mechanisms that can be easily explained to dog owners are the main limitations to the therapeutic use of FMT in canine clinical practices. Therefore, in order to facilitate the application of fecal transplant therapy in dogs, in this paper, we aim to review recent application of FMT in canine clinical treatments, as well as possible mechanisms that are involved in the process of the therapeutic effect of FMT.

## 2. Major Factors Influencing Canine Gut Microbiota

There are numerous factors that can cause alterations in canine gut microbiota (Figure 1). Such factors include diet [38], diseases [33], and medical interventions [39,40,41,42].

According to previous reports, other factors including obesity and diabetes [43,44], neurological disorders [45], breed, age, sex, living environment [46,47], and pregnancy also may have the capability to disrupt canine intestinal microbiota.

After the canine gut microbiota is disrupted by such elements, diversity and richness of normal microbial community reduces significantly, especially keystone bacteria that are responsible for providing a shield against pathogenic agents by forming colonization-resistant barriers. Changes in normal gut microbiota composition leads to impaired barrier protection; without this protection, disease-causing pathogenic agents are able to colonize the intestinal tube, grow, and cause pathological reactions.

## 3. Fecal Microbiota Transplantation in Dogs

FMT is an emerging non-pharmacological medical experimental treatment, which aims to restore intestinal microbial diversity and richness to a normal functional status. It is characterized as the transfer of fecal materials from a healthy donor to a diseased patient’s gastrointestinal tract. It has gained popularity in recent years. Nevertheless, studies on FMT application in canine clinical medicine are limited [48] (Table 1), however promising. This treatment, by far, has been used for intestine-originated disease in dogs when they do not respond to common therapy.

FMT has better long-term effects compared to standard treatments. For example, according to a previous study [49], when non-infectious acute diarrhea in adult dogs was treated with a single FMT administered by enema, or 7 days of oral metronidazole, although fecal consistency with both treatments was improved within 1 week, 4 weeks later, FMT treated dogs had firmer stools than the antibiotic-administered dogs. The fecal dysbiosis index (FDI) is a value computed based on qPCR results of eight bacterial taxa in the feces, which can offer a trustworthy clue as to changes in fecal microbiota composition [50]. FMT treatment effectively stabilizes FDI after 7 days, and remained in the normal range at day 28 in most dogs. However, for most dogs that were treated with metronidazole, FDI did not return to normal. In some cases, repeated FMT is required to achieve better therapeutic outcomes. When treating canine acute hemorrhagic diarrhea syndrome in eight dogs, single FMT did not have any clinical beneficial effect [51]. Although FMT results in quicker recovery, it may not upturn overall treatment efficiency in viral diseases. In one study, 66 puppies that had diarrhea associated with parvovirus infection were treated with either standard treatment or standard plus FMT [52]. The results showed that FMT is associated with faster resolution of diarrhea and shorter recovery period, but it did not increase survival rate compared to standard treatment groups.

FMT has also been used for treating IBD in dogs, and showed favorable results. A 10-year-old, neutered, male, 4-kg toy poodle with a prolonged history of vomiting and diarrhea, was treated with FMT [53]. After treatment, the patient’s clinical symptoms were improved, and no adverse effects were observed. Fecal microbiota analysis revealed that the recipient’s fecal microbiota resembled the healthy donor’s fecal microbial community, especially its diversity. Changes in gut microbial diversity due to a decrease in a specific genus in the gut microbiota may result in IBD in dogs. For example, a study involving nine IBD dogs showed that the proportion of *Fusobacterium* in the post-FMT fecal microbiome was significantly increased compared to pre-FMT [54]. Administration route is considered to be an important part of the FMT, and determines its efficacy. However, a study conducted with 16 idiopathic IBD dogs revealed that FMT treatment was both effective via oral and endoscopic methods [55]. However, more studies involving larger numbers of animals are needed for further evaluating the efficacy of different administration routes.

FMT has become popular for its high efficiency and safety for treating CDI in human medicine when the patient does not respond to antibiotic treatment or develops recurrent CDI (rCDI). Similarly, FMT is used for treating CDI patients in canine clinical practices. An 8-month-old, intact male French bulldog was presented with a 4-month history of intermittent large bowel diarrhea due to CDI [37]. The dog was treated with oral FMT that was obtained from a healthy beagle. After 2–3 days of treatment, stool consistency and frequency, and fecal blood and mucus became normal, and real-time PCR analysis and immunochromatography were negative for *C. difficile* antigen, toxin A&B genes and proteins. No adverse events were observed. In the meantime, other FMT administration routes, such as colonoscopy, also showed successful CDI treatment efficacy in other dogs [37,56].

## 4. Mechanisms of Fecal Microbiota Transplantation in Dogs

FMT aims to restore otherwise disrupted gut microbiota communities and transfer its compositional and functional status to normality. A healthy canine gut microbiome typically contains several different taxa and presents a high-level of taxonomic and functional diversity. Many of these taxa are innocuous and interact with each other to support host health and immune protection.

Although the exact mechanisms of FMT remain uncertain, there are four popular hypotheses: (1) niche exclusion, (2) increase competition for nutrition, (3) production of antimicrobials, and (4) increased production in secondary bile acids.

One of the potential mechanisms of FMT is competitive niche exclusion [58] (Figure 2). There are some situations when some fecal donor strains may compete for the same intestinal niches more successfully than the recipient’s pathogenic strains. Donor fecal materials can occupy these niches by excluding resident recipient microbial communities. FMT has been widely used for treating CDI in human medicine. For example, introduction of non-toxigenic *C. difficile* strains can reduce the recurrence of CDI in subjects [59].

During FMT treatment, donor fecal material not only directly interacts with the recipient’s native gut microbiota, but also performs indirect interactions, such as nutrition competition, with pathogens (Figure 3). Similar to niche exclusion, the mechanism of increased competition for nutrition is also aimed at reducing survival opportunities for pathogenic microbiomes, and possible mechanisms for FMT treatments in CDI patients.

Producing antimicrobials is another potential mechanism for FMT treatment [60] (Figure 4). This is also a competition-based strategy; interaction between host and donor gut microorganisms is the origin of bacteriocin production [61].

Lastly, the potential mechanism of FMT is increasing secondary bile acid production [62] (Figure 5). FMT can alter the recipient’s bile acid metabolism associated with alterations in gut microbiota composition [63].

In CDI treatment, for example, reduced primary bile acids and increased secondary bile acid production capacity has been reported after FMT. Meanwhile, it was also reported that FMT treatment can restore levels of bacteria of the Firmicutes phylum, and secondary bile acid metabolism, in CDI patients [64]. Similar to humans, in dogs, CDI is a known agent of acute diarrhea. Even though there are few articles that used FMT as a treatment of CDI in dogs, the mechanism of FMT treatment of CDI in human clinics is extensively studied. Its mechanism is a perfect example of a combination of niche exclusion and increased competition for nutrition, which are two of the main mechanisms of FMT treatment (Figure 6).

In a healthy gut, the microbial community typically contains numerous different taxa and presents very high taxonomic and functional diversity. Most of the taxa are harmless, and interact with one another to promote host health and immune protection. Healthy gut microbiota can establish an anticolonization barrier that is able to keep out opportunistic pathogens such as *Clostridium difficile* and *E. coli.* Although CDI is not a major concern in dogs, in human medicine, the mechanism of FMT is deeply studied based on CDI treatments. Antibiotic treatment results in lowered taxonomic diversity, further disrupting the gut microbiota, which allows the host to be colonized with *Clostridium difficile*. Spores of *Clostridium difficile* usually intersect dog gut microbiota through contaminated food or water intake. This leads to the development of CDI. Paradoxically, treatments for CDI in canines generally involve antibiotic interventions such as metronidazole [65]. Such antibiotics can exterminate *Clostridium difficile*, but spores can remain in the gut, which not only causes recurrence of CDI, but also leads to contaminations of soil, food, and water through feces. Transferring the fecal bacteria from a healthy donor to the patient’s gastrointestinal tract can restore the healthy gut microbiota, which can protect the host against CDI and other pathogenic inventions [37].

## 5. Risks and Limitations of Canine FMT

Although various studies support the notion that, in general, FMT is a safe approach for treating gastrointestinal diseases with little adverse events, safety concerns still remain the main limitation in fecal transplant therapy in canine medicine, coupled with lack of practical, reusable guidelines and proper regulations. Another major issue with FMT therapy is that it may be possible to transmit harmful bacteria that live in the donor without inducing clinical symptoms [66], such as drug-resistant *E. coli* bacteria. Therefore, a strict process for donor screening should be required, including clinical and biological examinations. There is no doubt that selecting a route for FMT administration is a key element for successful treatment; however, based on previous studies, it may not be as important as we thought. In most cases, various delivery methods showed similar therapeutic efficacies. Therefore, it may only depend on the skills and equipment the veterinarians possess and the choices the owner makes. However, we suggest the veterinary specialists offer the owners a detailed explanation of different FMT methods and their risks before the procedures.

As stated above, the canine gut microbiome is also composed of fungi and viruses. Without doubt, these also have an impact on FMT efficacy. In human medicine, it was concluded that fungi might potentially influence FMT efficacy in rCDI [67]. However, it is an unknown area in canine fecal transfer therapy, and more studies are needed.

Additionally, it should be made clear before treatment that FMT has other risks, such as transmission of pathogens, multidrug-resistant bacteria, and other adverse effects including diarrhea, constipation, or colic.

## 6. Future Perspective of Canine FMT

The future of FMT for treating gastrointestinal diseases in veterinary medicine will be bright, especially in canine clinical practices, as dogs are very sensitive to changes in gut microbial environment, and this usually results in problems such as diarrhea and colic. Such conditions can be ameliorated by restoring the disrupted gut microbiota community by fecal transplant therapy, with little adverse events.

Although some veterinarians have proposed steps for canine FMT based on research articles and experience, there is still much more room for improvement and exploration. For example, with the rapid development of 16S rRNA sequencing, which is becoming more widely available and less costly, we are now able to choose donors based on a specific disorder in the recipient dogs. For example, a previous study found that recipients of FMT showed an increasing relative abundance of *Fusobacteria* when treating dogs with IBD [53]. Therefore, transferring fecal matter that contains a relatively high abundance of *Fusobacteria* may be more effective when treating IBD in dogs. Future research should explore such relationships between a specific disease and changes in gut microbiota composition before and after FMT treatment, which may greatly enhance the efficacy of fecal transplant therapy.

Furthermore, it has been suggested that gut microbiota dysbiosis can induce mental stress by influencing the gut–brain axis.

Behavior abnormalities such as aggressiveness are most frequently observed in mentally stressed dogs that present anxiety disorders, which indicates imbalance in the gut microbial composition [68]. Indeed, transferring gut microbiota also transforms the mood of the donor. Upon transplanting fecal microbiota from patients with depression to the intestinal tract of microbiota-depleted rats, the rats showed anxiety-like behavior [69], suggesting that mental stress can be transmitted via gut microbiota. It was also reported that *Lactobacillus* spp. can improve social communication in stressed mice [70], and *Bacteroides* spp. ameliorated anxiety-like behaviors in mice, which may be achieved by the restoration of certain bacterial metabolites [71]. Therefore, gut microbiota could be a target in the treatment and prevention of abnormal behaviors in dogs. Establishing a stool bank for fecal materials from psychically and mentally sound donors for use in behavior rehabilitation could be a novel therapeutic approach for mentally stressed dogs, especially in older dogs.

The current donor screening process primarily aims to exclude pathogens to increase the safety of FMT, yet there is no widely accepted agreement on selecting donors based on parameters. In the future, the donor selection process should involve inspecting the microbial diversity and ideal *Bacteroidetes* versus *Firmicutes* ratio, which are the indicators of healthy gut microbiota. In this process, behavior evaluation is also recommended for donor candidates. At the same time, there is little doubt that donor selection criteria will be extended in the future if newly discovered pathogens are proven to be able to disrupt normal gut microbiota composition or transmit via FMT. Therefore, veterinary specialists should be more flexible, consider endemic diseases, and adjust the donor screening to a local characteristic approach by consulting with local departments of infectious diseases and veterinary clinical microbiologists.

Stool storage is also an important step in FMT. In dogs, currently, in most cases, freshly collected feces are used for FMT treatments. In human studies, it was found that frozen stool samples are just as effective as fresh fecal material [72]. This result has more significant meaning for veterinary clinics as using prescreened fecal materials in storage is a cost- and time-effective and much safer approach compared to fresh FMT. Frozen stool can overcome the geographic limitation and facilitate increased implementation of FMT in canine clinical practices (Figure 7).

Therefore, establishing a canine stool bank may be a good idea if an intensive screening process is involved. However, some critical issues should be taken into consideration when building stool banks. Firstly, more strict screening and selecting of donor dogs should be conducted based on stool microbial culture, antibiotic resistance testing, and necessary hematological examination to minimize the risk factors. Stool banks may not only save time and facilitate the application of FMT in canine clinics and ranches with simple procedures, but also reduce costs for donor or stool handling. Furthermore, stool bank donors are required for detailed information registration [73]; therefore, it is easy to track the donor during and after the FMT procedure, and further monitor and ensure safety of samples in the stool bank.

Meanwhile, obesity is becoming a major health concern in dogs, and it is related to metabolic abnormalities [46]. It has been shown that gut microbiota can change in overweight horses after losing weight [74,75]; specifically, the diversity of the fecal microbiota showed a significant increase after losing weight. Given the above results and the gut microbiomes’ role in fitness, it is highly possible that one could select a lean dogs’ fecal material according to body condition score, and use this as a means of treatment in overweight dogs for weight loss, which would be a safe and economical approach.

In addition to the importance of the donor fecal contents, studies have suggested that the gut microbiome composition of the recipient also plays an important role in their clinical response to FMT [76,77]. Thus, antibiotic pretreatment has been effectively used to alter resident gut microbiota in recipient patients to increase FMT efficacy by eliminating the potential competitive advantage of existing microorganisms that may decrease the colonization by microbes present in the transplanted feces. Meta-analyses of human and mice studies also support the notion that antibiotic pretreatment enhances FMT efficiency [78,79]. This process is based on the ‘ecological niche’ hypothesis, which states that unsuccessful FMT occurs because of the ‘barrier’ effect created by the recipient’s gut microbiota community against the colonization of the donor microbial population by creating competition for ecological niches.

Antibiotic treatment prior to FMT has never been studied in dogs. Recently, however, emerging evidence has suggested that the usage of antibiotics in animals is associated with many adverse effects. Among them, antibiotic-resistant bacteria are the biggest problem, because, unlike farm animals, dogs are adored for their companionship, especially in children. It could be very dangerous if such bacteria were transmitted from dogs to young children, elderly individuals, or other adults with immunodeficiency diseases. Therefore, we do not recommend using antibiotics as pretreatment agents when conducting FMT in client-owned dogs.

Laxatives are an alternative option for ‘getting rid of’ recipient gut microbiota, and are gaining more and more popularity in human FMT treatments considering that antibiotic treatment has the potential risk of causing multidrug resistance, which is also one of the major concerns and limitations for wider FMT implementations in clinical practices. A study showed that 40 mL/kg of polyethylene glycol (PEG 4000) provides efficient bowel cleaning in human subjects [80]. Another recent study reported that 425 g/L of PEG administered by oral gastric gavages at 20 min intervals can empty the intestine and decrease the gut microbiota by 90% after four successive administrations in mice [81]. In addition to mice and human studies, an article reported that stomach tube administration of 8 mg/kg of PEG was effective in dogs for bowel cleaning [82]. It suggested that PEG was equal to, or slightly more effective than senna when used for intestine fecal cleaning before radiology and colonoscopy. Although this study concluded that there were no adverse effects, repeated stomach tube insertion with general anesthesia may be dangerous in elderly or anesthesia-allergic dogs.

In conclusion, even though it is quite possible to achieve successful FMT treatment in dogs without cleaning their gut microbiota [37,83], it is advisable that veterinary specialists should pretreat recipient dogs with laxatives such as PEG. This increases FMT efficacy by reducing repeated treatment, which is a huge improvement in canine welfare if FMT is delivered through other routes, such as colonoscopy and enema.

Additionally, poop pills, freeze-dried encapsulated stool products, are widely accessible for human patients, and show very promising results [72,84,85]. For example, when 49 CDI patients were treated with encapsulated FMT, the success rate was 88%. However, there is only one report that used commercially available, encapsulated fecal materials used in canine research [83]. Therefore, there may be a prospective study on this matter.

## 7. Conclusions

FMT is a good therapeutic choice for treating gut-microbiota-related disorders in canine clinics, and has been used for gastrointestinal disorders, including CDI and IBD, and is now becoming accepted as a treatment option by medical specialists around the world. Successful FMT is heavily dependent on the selection of the most suitable donor, as the ideal content of fecal material plays a critical role as a regulator of the disrupted gut microbiota community in the recipient. In dogs, FMT still is at an early stage, with increased use in clinical settings, regulations, and standardization becoming an urgent matter. Therefore, in this paper, we described the current application of FMT therapy in dogs and the possible mechanisms that are involved in the process, as well as future prospects regarding clinical techniques, to facilitate its therapeutic usage.

## Figures and Tables

**Figure 1 vetsci-09-00396-f001:**
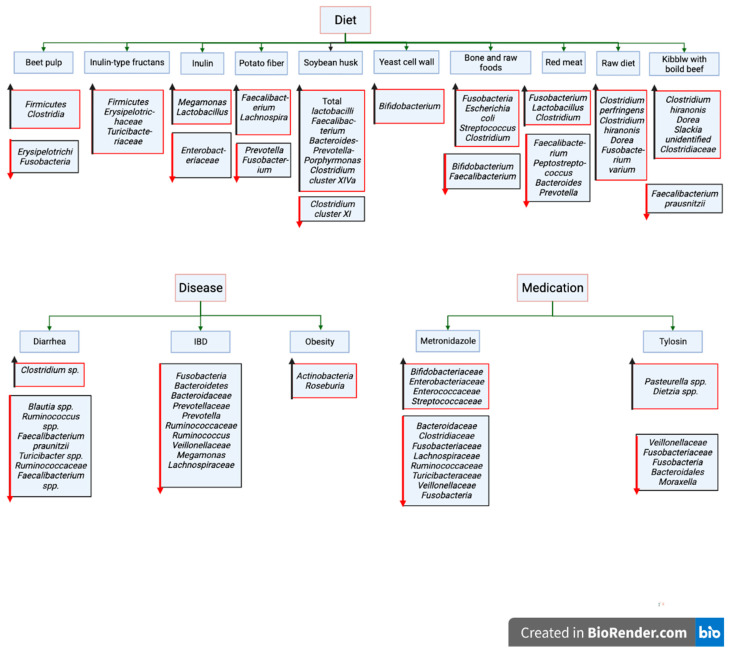
Major influencing factors of canine gut microbiota. (Arrows: upward, increased relative abundance; downward, decreased relative abundance). (Created with BioRender.com, accessed date 18 June 2022).

**Figure 2 vetsci-09-00396-f002:**
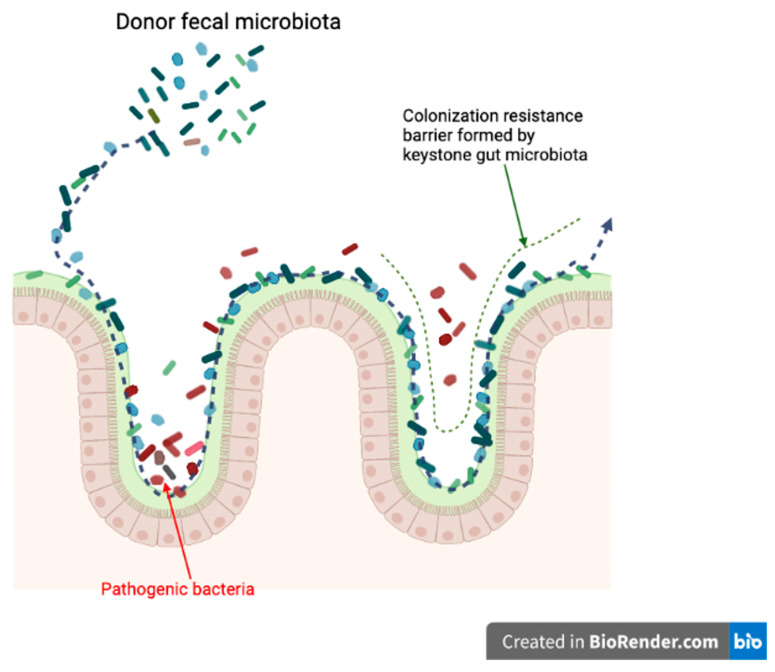
Potential mechanism of FMT; niche exclusion. (Created with BioRender.com, accessed date 18 June 2022).

**Figure 3 vetsci-09-00396-f003:**
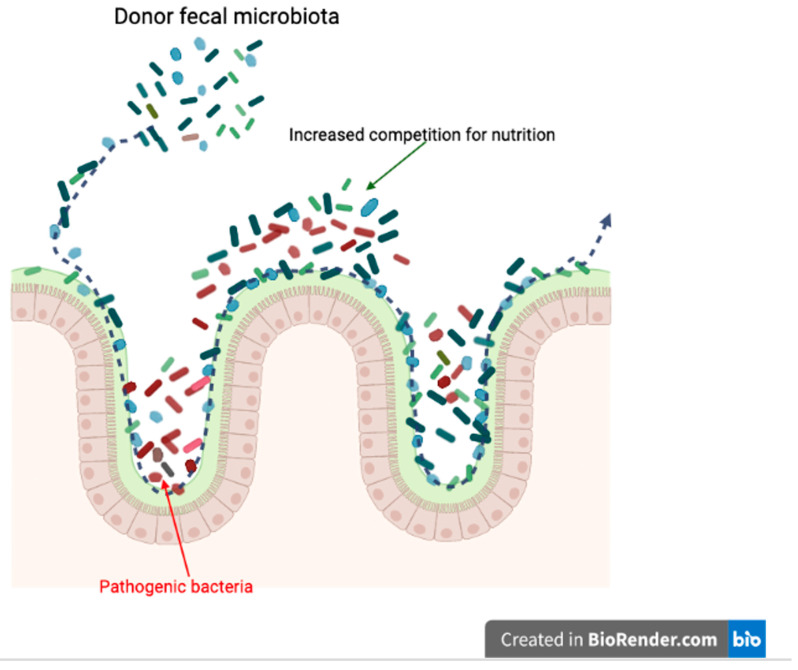
Potential mechanism of FMT; increased competition for nutrition. (Created with BioRender.com, accessed date 18 June 2022).

**Figure 4 vetsci-09-00396-f004:**
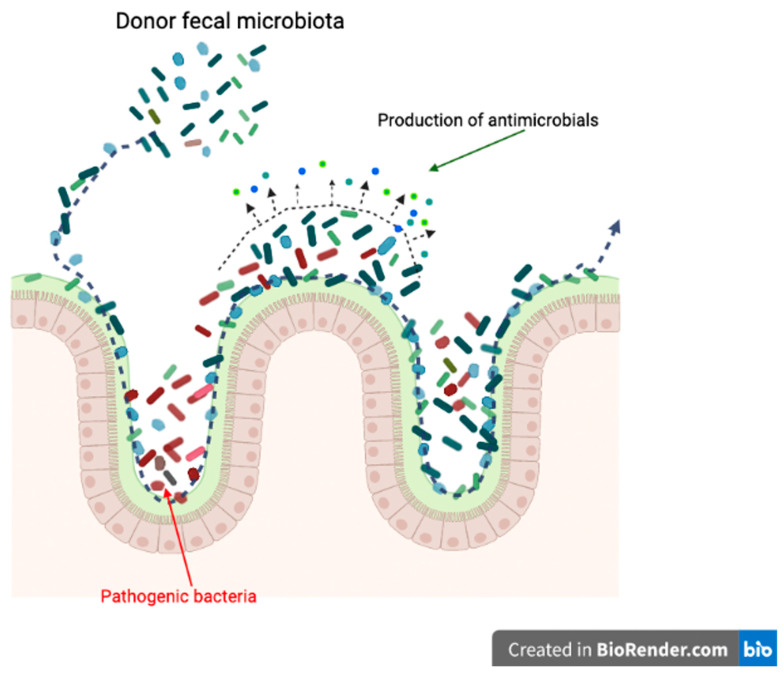
Potential mechanism of FMT; production of antimicrobials. (Created with BioRender.com, accessed date 18 June 2022).

**Figure 5 vetsci-09-00396-f005:**
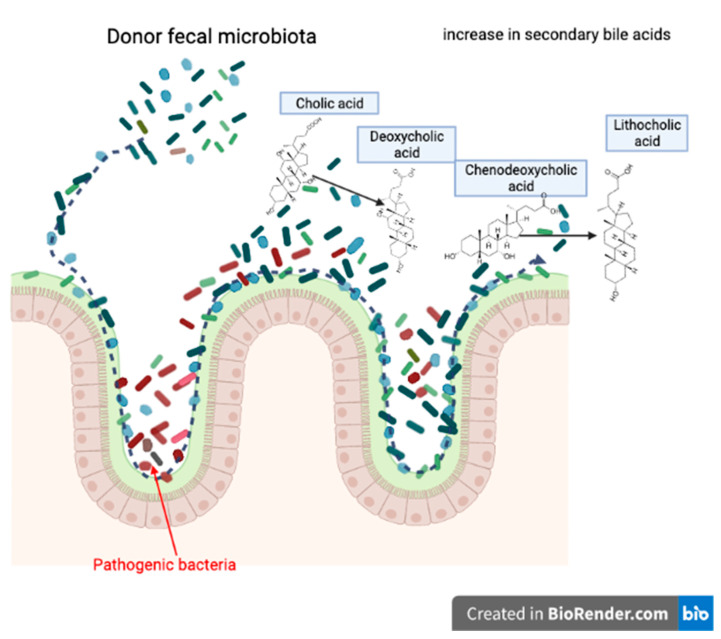
Potential mechanism of FMT; increased production of secondary bile acids. (Created with BioRender.com, accessed date 18 June 2022).

**Figure 6 vetsci-09-00396-f006:**
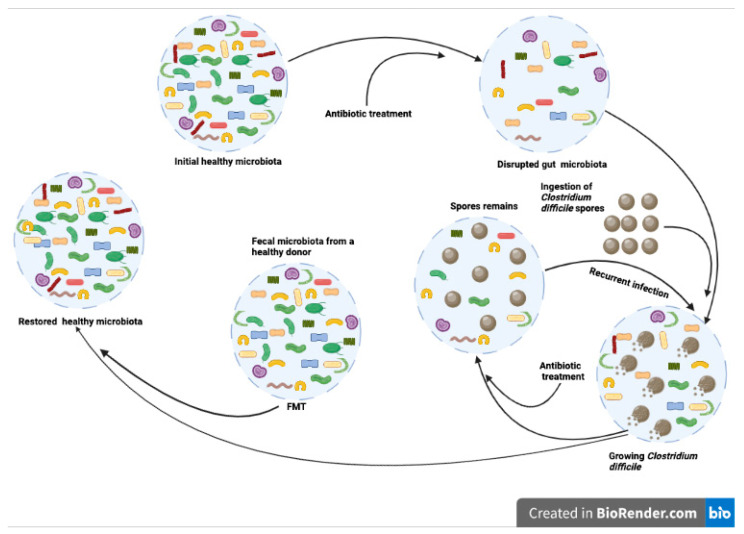
Possible mechanism of FMT therapy for *Clostridium difficile* infection. (Created with BioRender.com, accessed date 18 June 2022).

**Figure 7 vetsci-09-00396-f007:**
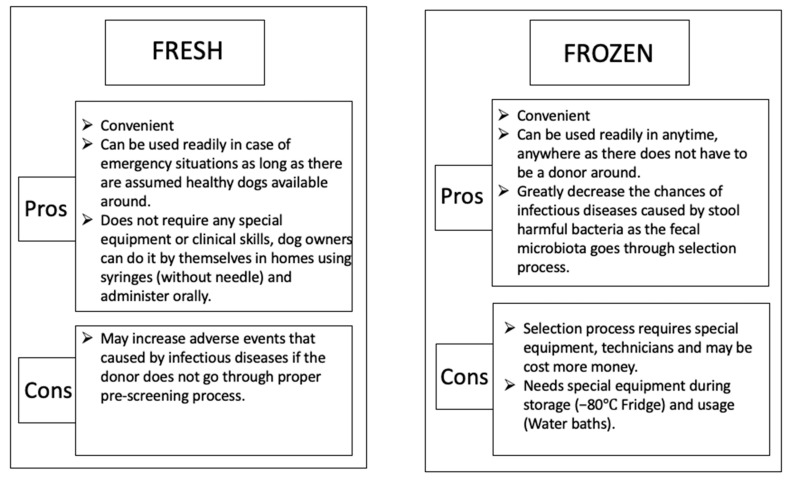
Pros and cons of preparing fresh and frozen stools for canine FMT treatment.

**Table 1 vetsci-09-00396-t001:** Peer-reviewed literature of canine fecal microbiota transplantation.

Author, Year	Recipient Feature	Number of Dogs	Frequency of FMT	Delivery Route	Clinical Effects	Effects on Fecal Microbiota	Method for Fecal Preparation
Burton et al., 2016 [57]	Weaning puppies, postweaning diarrhea	11 FMT 12 controls	5 days, once per day	Oral	No difference in fecal consistency between FMT and control puppies	Wide variability of microbiome in puppies, no clustering with donor microbiome observed	10 mL fecal suspension (100 g pooled dam feces mixed with 200 mL 2% fat cow’s milk after filtration)
Bottero et al., 2017 [55]	IBD refractory to conventional treatment	16 adult dogs with severe, refractory IBD of >1 year duration	Oral treatment group received FMT q 48–72 h	9 dogs endoscopy + oral, 7 dogs oral	Overall, mean CCECAI seemed to decrease in most dogs following FMT. Heterogeneous clinical presentation and concurrent treatments complicate evaluation	Not applicable	60–80 g feces for dogs <20 kg BW, 100–150 g for dogs > 20 kg BW. 1:1 dilution with 0.9% saline, filtered and mixed with low-fat yogurt as enrichment solution
Pereia et al., 2018 [52]	Parvovirus infection	33 received standard treatment, 33 received FMT in addition	FMT administered within 5–12 h of admission and q 48 h thereafter	Endoscopy	No difference in mortality rate, FMT dogs had quicker resolution of diarrhea, and shorter hospitalization	Not applicable	10 g feces administered per puppy. 1:1 dilution with saline
Nina et al., 2019 [53]	IBD refractory to antibiotic and immunosuppressive treatment over time	10-year-old toy poodle	9 treatments within 6 months	Endoscopy	Improved CIBDAI.	Increased in *Fusobacteria*, *Firmicutes* and *Bacteroidetes*, decreased in *Proteobacteria*. Clustered phylogenetically with donor	Feces diluted 1:3 with ringer lactate. The dog received approximately 3 g feces/kg body weight
Sugita et al., 2019 [37]	Intermittent large bowel diarrhea, 4 months of duration, feces positive for CD (PCR and toxins A & B)	8-month-old French bulldog		Oral	Normalization of fecal consistency and defecation frequency within 2–3 days, without recurrence of CD or diarrhea over 190 days	Not applicable	30 mL fecal suspension (60 g feces diluted in 50 mL tap water after filtration) given orally. Equivalent to approximately 2.5–3 g feces/kg BW
Chitman et al., 2020 [49]	Uncomplicated acute diarrhea of <14 days duration		11 dogs received a single FMT, 7 dogs received metronidazole 15 mg/kg q 12 h for 7 days	Endoscopy	Lower (better) fecal score at days 7 and 28 for both treatments, FMT fecal score lower than metronidazole at day 28	Fecal dysbiosis indexes better with FMT than metronidazole at days 7 and 28. FMT dogs tended to cluster healthy dogs at day 28, unlike metronidazole dogs	Fresh feces mixed with 60 mL 0.9% NaCl in a blender. Blend on high until the stool is liquefied and no larger pieces are seen. For very large dogs a larger volume of saline may be needed to obtain sufficiently liquefied fecal solution
Diniz et al., 2021 [56]	Chronic-recurring pasty large bowel diarrhea	4-year-old female golden retriever	Received FMT via colonoscopy	Colonoscopy	*Clostridium difficile* no longer present in the dog’s stool	Not applicable	Approximately 65 g of feces were diluted in 250 mL of sterilized PBS
Gal et al., 2021 [51]	Canine acute hemorrhagic diarrhea syndrome	8 dogs aged 3–12 years old	Received FMT via colonoscopy	Colonoscopy	There were no significant differences in median AHDS clinical scores between FMT-recipients and sham-treated controls	Increased microbiota diversity. Short-chain fatty acid producers including *Eubacterium biforme*, *Faecalibacterium prausnitzii*, and *Prevotella copri* were significantly decreased	Stool was homogenized at room temperature in a sterilized blender at a ratio of 1-part stool/4 parts saline. The suspension was passed through a sterilized sieve to remove large particles

## Data Availability

Not applicable.

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
