# Peer review of "Canine Fecal Microbiota Transplantation: Current Application and Possible Mechanisms"

_vetsci, 2022, doi:10.3390/vetsci9080396_

Round 1
Reviewer 1 Report
Thank you for your paper,
Good summary of pros and cons. Very detailed. Good use of graphic materials, flow charts and tables.
Inappropriate italics in text e.g lines 41, 58, 82.
Suggestion: selecting for particular traits potentially problematical – select for suitable body condition score and you may select for other problems - behavioural problems, susceptibility to different diseases. How big would be your list of traits from which to select?
line 110 In some cases, repeated FMT is required to achieve better therapeutic outcomes. Does author have a reference for this?
Line 127 Route of administration: intuitively, per-rectal administration would be expected to be far more effective than oral route. A study involving 16 cases and no controls cannot reliably contradict this.
Need more studies involving large number of animals.
line 149 mechanisms (plural) may be involveld
line 204 - good discussion of risks and limitations
Any thoughts on importance of diet in both recipient and donor, eg raw diet in achieving healthy microbiota. Would you recommend donor animals are/are not raw fed?
This paper achieves its aims to describe FMT in dogs, and explore possible mechanisms

Author Response
Good summary of pros and cons. Very detailed. Good use of graphic materials, flow charts and tables.
Inappropriate italics in text e.g lines 41, 58, 82.
Dear Reviewer, thank you for your comment. We changed the inappropriate italics as you suggested.
Suggestion: selecting for particular traits potentially problematical – select for suitable body condition score and you may select for other problems - behavioural problems, susceptibility to different diseases. How big would be your list of traits from which to select?
Dear Reviewer, thank you for your comment. If we focus on one problem, others may be neglected when selecting donors. Therefore, we tried to include as many as possible. As we see in clinical practices, many veterinarians only care about if a donor dog is healthy or not physically, and neglect the behavior issues. We included theses traits, because we hope they could serve as a reminder for veterinary specialist when selecting donors.
line 110 In some cases, repeated FMT is required to achieve better therapeutic outcomes. Does author have a reference for this?
Dear Reviewer, thank you for your comment. Yes, sorry, we have a reference for this. ‘In some cases, repeated FMT is required to achieve better therapeutic outcomes. When treating canine acute hemorrhagic diarrhea syndrome in 8 dogs, single FMT did not have any clinical beneficial effect[51].’. reference 51 is a reference for this statement.
Line 127 Route of administration: intuitively, per-rectal administration would be expected to be far more effective than oral route. A study involving 16 cases and no controls cannot reliably contradict this.
Need more studies involving large number of animals.
Dear Reviewer, thank you for your comment. Yes, you are right. In human clinics per-rectal administration of FMT is proven to be more effective. But, in dogs, we only found this one study. And as you suggested, we added the sentence ‘However, more studies involving larger number of animals are needed for further evaluating the efficacy of different administration routes.’
line 149 mechanisms (plural) may be involveld
Dear Reviewer, thank you for your comment. We corrected the sentence with plural ‘Although the exact mechanisms of FMT are still remaining imprecise,……..’
line 204 - good discussion of risks and limitations
Any thoughts on importance of diet in both recipient and donor, eg raw diet in achieving healthy microbiota. Would you recommend donor animals are/are not raw fed?
Dear Reviewer, thank you for your comment. Although studies have shown that raw diets can alter canine gut microbiota, but we do not recommend changing diets in either donor or recipient dogs. Diets have very dramatic and sudden effect on gut microbiota, changing diets may cause unexpected problems. Therefore, we do not recommend changing to raw diet or any other diets during FMT treatment in dogs who donate or receive feces.
This paper achieves its aims to describe FMT in dogs, and explore possible mechanisms.
Dear Reviewer, thank you for your comment.

Reviewer 2 Report
The manuscript entitled "Canine faecal microbiota transplantation- current application 2 and possible mechanisms" presents a very actual work in clinical practice. Overall is very well written and present, but are necessary corrections:
- Abstract: needs a conclusion
- Figure 1 bacteria names should be in italic
- table 1 references should not be like Burton et al,[57] 2016, they should be Burton et al, 2016 [57]
-table 1 0verall to overall an sentence should start with maiuscule
- Figure 2 - 5: is the figure from the authors? if not should be mentioned where the image was removed
- Subtitles figures: sentence starts with maiuscule
- Figure 6: name bacteria in italic
- Risks and limitations of canine FMT - why it is in yellow?
-read the text and all names in Latin, from bacteria should be in italic, in some sections they are not
- conclusion: is not necessary to repeat the aim of the work, in this section with should be given conclusion and aims for the future in practice and for the technique
- Institutional Review Board Statement, Informed Consent Statement, Acknowledgments and appendix: eliminate this section since it is a review
- references: need correction some data is missing in some of them and they don´t seem to be formatted according to the guidelines of the journal
Author Response
Abstract: needs a conclusion
Dear Reviewer, thank you for your comment. We made a conclusion as you suggested.
- Figure 1 bacteria names should be in italic
Dear Reviewer, thank you for your comment. We corrected the style of bacteria names in Figure 1 as you suggested.
- table 1 references should not be like Burton et al,[57] 2016, they should be Burton et al, 2016 [57]
Dear Reviewer, thank you for your comment. We corrected Table 1 references according to you suggestion.
-table 1 0verall to overall an sentence should start with maiuscule
Dear Reviewer, thank you for your comment. We corrected is as you suggested.
- Figure 2 - 5: is the figure from the authors? if not should be mentioned where the image was removed
Dear Reviewer, thank you for your comment. All images are made by the author using BioRender.com
- Subtitles figures: sentence starts with maiuscule
Dear Reviewer, thank you for your comment. We corrected the problem with subtitle figures.
- Figure 6: name bacteria in italic
Dear Reviewer, thank you for your comment. We corrected the style of bacteria names as you suggested.
- Risks and limitations of canine FMT - why it is in yellow?
Dear Reviewer, thank you for your comment. It was a note to another author, we forgot to remove. We removed yellow.
-read the text and all names in Latin, from bacteria should be in italic, in some sections they are not
Dear Reviewer, thank you for your comment. We corrected all bacteria names as you suggested.
- conclusion: is not necessary to repeat the aim of the work, in this section with should be given conclusion and aims for the future in practice and for the technique
Dear Reviewer, thank you for your comment. We revised the last sentence as you suggested: Therefore, in this paper we aimed to describe the current application of FMT therapy in dogs and possible mechanisms that are involved in the process, as well as future prospective regarding clinical techniques, to facilitate its therapeutic usage.
- Institutional Review Board Statement, Informed Consent Statement, Acknowledgments and appendix: eliminate this section since it is a review
Dear Reviewer, thank you for your comment. We deleted this information.
- references: need correction some data is missing in some of them and they don´t seem to be formatted according to the guidelines of the journal
Dear Reviewer, thank you for your comment. We tried both ‘Veterinary Sciences’ and ‘MDPI’ reference styles, however, the DOIs are still missing. Therefore, we manually added all DOIs to the references as you suggested.
